health and disease and epidemiology/theoretical biology/applied mathematics

COVID-19, modelling, basic reproduction number, immunity, prevention

**Author for correspondence:**
Tom Britton
e-mail: tom.britton@math.su.se

# The risk for a new COVID-19 wave and how it depends on $R_0$, the current immunity level and current restrictions

Tom Britton[1], Pieter Trapman[1] and Frank Ball[2]

[1]Department of Mathematics, Stockholm University, Stockholm, Sweden
[2]School of Mathematical Sciences, University of Nottingham, Nottingham, UK

TB, 0000-0002-9228-7357; PT, 0000-0003-0569-1659

The COVID-19 pandemic has hit different regions differently. The current disease-induced immunity level $\hat{i}$ in a region approximately equals the cumulative fraction infected, which primarily depends on two factors: (i) the initial potential for COVID-19 in the region ($R_0$), and (ii) the preventive measures put in place. Using a mathematical model including heterogeneities owing to age, social activity and susceptibility, and allowing for time-varying preventive measures, the risk for a new epidemic wave and its doubling time are investigated. Focus lies on quantifying the minimal overall effect of preventive measures $p_{\text{Min}}$ needed to prevent a future outbreak. It is shown that $\hat{i}$ plays a more influential roll than when immunity is obtained from vaccination. Secondly, by comparing regions with different $R_0$ and $\hat{i}$ it is shown that regions with lower $R_0$ and low $\hat{i}$ may need higher preventive measures ($p_{\text{Min}}$) compared with regions having higher $R_0$ but also higher $\hat{i}$, even when such immunity levels are far from herd immunity. Our results are illustrated on different regions but these comparisons contain lots of uncertainty due to simplistic model assumptions and insufficient data fitting, and should accordingly be interpreted with caution.

## 1. Introduction

At the end of 2020, prior to vaccination, COVID-19 was spreading in many parts of the world. In some regions spreading was on decline, most often owing to the implementation of effective preventive measures, whereas in others it was on the rise again. Up until the time when a large community fraction has been vaccinated (achieved in mid-2021 in many western countries, but not yet in most of the world) regions where current spreading is low face two competing interests: lifting restrictions

to normalize society, and to keep (or even strengthen) restrictions in order to avoid a new major COVID-19 outbreak. The minimal overall effect of preventive measures $p_{\text{Min}}$ needed to avoid a large future outbreak (or to make it start declining in the case of an ongoing outbreak) and how it depends on the basic reproduction number $R_0$ and the current immunity level $\hat{\imath}$, are hence highly important questions which are investigated here. In addition, we consider the doubling time of a new outbreak should one take place, which gives an indication of its impact before additional preventive measures would be implemented.

In [1], mathematical arguments were used to claim that the disease-induced herd immunity threshold is substantially lower than the vaccine-induced (i.e. classical) herd immunity threshold. However, the vast majority of regions, possibly excluding sparsely populated regions which might even have $R_0 < 1$, are not even close to the herd immunity level. In the present paper, we hence focus on immunity levels (often substantially) lower than the herd immunity level. Such immunity might still give some protection and here we aim to quantify broadly this effect and study how it depends on whether it is obtained by disease spreading or by vaccination.

The basic reproduction number $R_0$ quantifies the initial potential of an epidemic outbreak for a particular disease (or disease variant) in a particular region, and is defined as the *average* number of new infections caused by a *typical* infected individual in the beginning of the epidemic outbreak (before preventive measures are put in place and before population immunity starts to build up), [2]. For COVID-19, estimates of $R_0$ vary substantially between different regions, e.g. between 2 and 5 among 11 European countries for the original strain of COVID-19 [3].

Preventive measures aim to reduce the average number of infections caused by an infective, by either reducing the risk of transmission given a contact (e.g. hand washing, wearing face mask), reducing the number of daily contacts (e.g. physical distancing, school closure) and/or reducing the effective infectious period (e.g. testing and isolating, treatment). Let $p(t)$ denote the overall effect of such preventive measures at time $t$, where $0 \leq p(t) \leq 1$, and with $p(t) = 0$ corresponding to no preventive measures and $p(t) \approx 1$ meaning more or less complete isolation of all individuals.

Let $\hat{\imath}(t)$ denote the community fraction that cannot get infected at time $t$, a few of these being currently infectious, but the majority having recovered from the disease and are now immune. At time $t$, it is the current (or effective) reproduction number $R_t$ of a region that determines if a new main outbreak can take place or not. In particular, a region with low current transmission avoids the risk for a large new outbreak as long as $R_t < 1$, and regions with ongoing transmission will see a rise in transmission if $R_t > 1$ and a decline in transmission whenever $R_t < 1$.

For simple epidemic models, which assume a homogeneous community that mixes homogeneously, it is well known that $R_t = R_0(1 - p(t))(1 - \hat{\imath}(t))$, since $R_0$ is reduced both due to the preventive measures and from the fact that some contacts will be with already infected people. This implies that $R_t \leq 1$ is equivalent to $p(t) \geq 1 - 1/(R_0(1 - \hat{\imath}(t)))$, thus quantifying, in terms of $R_0$ and the current immunity level $\hat{\imath}(t)$, the minimal amount of preventive measures needed to avoid a new large outbreak.

For more realistic epidemic models, this simple relation between $R_t$ and $R_0$, $p(t)$ and $\hat{\imath}(t)$ does not hold. In fact, for many epidemic models acknowledging population heterogeneities it holds that $R_t < R_0(1 - p(t))(1 - \hat{\imath}(t))$. The main reason is that individuals having high social activity and/or high susceptibility are more likely to be infected early in the epidemic, implying that individuals at risk later in the epidemic will on average be less susceptible and less socially active, thus also infecting fewer if they become infected [1]. Here we extend the epidemic model studied in [1], in which empirical age-structured social mixing is combined with independent variable social activity within age groups, to now also allow for variable susceptibility and infectivity within age groups. (The mixing by age contact rate matrix is given in the electronic supplementary material, where it is denoted by $A^{\dagger}$. It is multiplied by a constant to yield an epidemic having the desired $R_0$.) For this model, the aim is to quantify $R_t$ as a function of $R_0$, $p(t)$ and $\hat{\imath}(t)$, and in particular to quantify the minimal amount of restrictions $p_{\text{Min}}$ for a region having initial basic reproduction number $R_0$ and current immunity level $\hat{\imath}$ (the index $t$ is now dropped and implicitly considered as current time).

We illustrate our findings by expressing $p_{\text{Min}}$ and the doubling time for different regions in Europe and the USA, but these illustrations are by no means exact. First, the model is clearly a simplification of the ongoing COVID-19 pandemic, but even more so the estimates of $R_0$ and the current immunity level $\hat{\imath}$ for different regions contain appreciable uncertainty. Second, we assume that preventive measures are applied uniformly in the population, whereas in reality they are likely to be targeted at

certain groups, such as the elderly. Nevertheless our results allow, for the first time to our knowledge, a risk comparison between regions having different $R_0$ and different immunity level $\hat{i}$.

## 2. An epidemic model with age cohorts, variable social activity and variable susceptibility

The epidemic model is based on the SEIR epidemic model in Britton *et al.* [1]. Individuals are divided into six different age groups, and mixing patterns are taken from the empirical study of Wallinga *et al.* [4]. Social activity is modelled in a simplistic way not claimed to mimic reality, but still allowing for variable social activity which is often neglected. Within each age group individuals are divided into three categories: 50% have normal social activity, 25% have low social activity (half as many contacts as those with normal activity) and 25% have high social activity (double activity). It is important to stress that social activity affects both the risk of getting infected as well as infecting others in that socially active individuals have more contacts both when susceptible and when being infectious.

To this model with age cohorts and variable social activity, studied in [1], we now also add variable susceptibility [5], which is done similarly to variable social activity. We assume that 50% have normal susceptibility, 25% have half that susceptibility and 25% are twice as susceptible, and the variable susceptibility is assumed to be independent of both social activity level and age group. As a consequence, our model does not take into account e.g. lower susceptibility among elderly people although this is partly captured by the lower contact rates for this age group. The choice to divide social activity as well as susceptibility into three groups as above is of course arbitrary. This choice of heterogeneity structure is quite moderate in that there is no tail (with individuals having very high or low social activity and/or susceptibility) and the coefficient of variation equals 0.48 which is moderate (see the electronic supplementary material for further comments).

It seems natural to also add variable infectivity for individuals who become infected. However, in the electronic supplementary material we show that such variable infectivity would have no effect on our results, provided the infectivity is assumed to be independent of age, susceptibility and social activity. For this reason, we have omitted variable infectivity in our model: a model where also infectivity varies (independent of age, susceptibility and social activity) would give identical results.

We use a deterministic SEIR epidemic model (see electronic supplementary material) with a total of $6 \times 3 \times 3 = 54$ different types of individuals, but very similar results would be obtained from simulations of a corresponding stochastic model assuming a large population (which can be proved using methods in [6], Part I, theorem 2.2.7). The latent state 'E' (for exposed) is assumed to have mean 3 days followed by an infectious period ('I') having mean 4 days, thus being quite close to other models for the spread of COVID-19 [3] (the estimates are based on data from the initial outbreak in Wuhan but are believed to apply in general since latent and infectious periods prior to prevention are mainly biologically driven). Details of the model are given in the electronic supplementary material, where it is explained that, apart from doubling times, our results hold also for the corresponding model in which the latent and infectious periods are not necessarily exponentially distributed, and more generally for the model in which infectives have independent and identically distributed infectivity profiles.

It is straightforward to numerically derive properties of the model, such as the basic reproduction number $R_0$, the time dynamics and its final fraction infected when the epidemic stops (see electronic supplementary material for further details).

## 3. Prevention

During the outbreak(s), preventive measures of varying magnitude may be put in place. We assume that these preventive measures do not affect the latent and infectious periods, but only that they reduce the rate of infectious contacts. More precisely, we make the strong and somewhat restrictive assumption that, at time $t$, all contacts (between the 54 types of individuals) are reduced by the *same* factor $p(t)$. This assumption can easily be relaxed, but to explore all possibilities of contact reduction is infeasible, and among all specific preventions the uniform one, where all contacts are reduced by the same factor, is the most natural choice. Even when assuming such uniform reduction of contact rates, its reduction may vary in time in different ways. However, in the electronic supplementary material it is shown that for a fixed overall epidemic size the exact time allocation of the preventive measures has

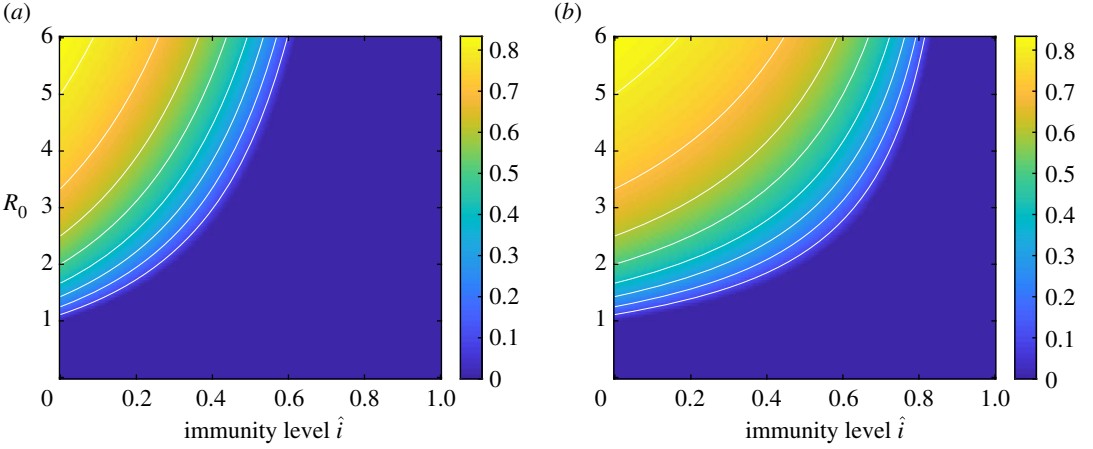

**Figure 1.** Plot of the minimal amount of preventive measures, $p_{\text{Min}}$, necessary to avoid a new large outbreak, as a function of $R_0$ and the current immunity level $\hat{\imath}$. The left plot is for disease-induced immunity and the right plot is for vaccine-induced immunity.

negligible effect in our model: any time-varying preventive measures $\{p(t); 0 \leq t \leq t_0\}$ from the start of the epidemic up until some fixed time $t_0$, leading to the same overall fraction infected, will have nearly the same fractions infected among the different types of individuals (see electronic supplementary material for details). Thus early mild preventive measures will result in the same composition of infected individuals as doing nothing and then suddenly going to a full lockdown, assuming the two preventive measures lead to the same overall fraction infected.

## 4. The minimal amount of preventive measures $p_{\text{Min}}$

Consider a large community in which COVID-19 has spread in one or several waves according to our model with some fixed value of $R_0$, and for which preventive measures $\{p(t); 0 \leq t \leq t_0\}$ were put in place (with the same preventive effect on all type of contacts). We further assume that at time $t_0$ the transmission has more or less stopped, with a fraction $\hat{\imath}$ having been infected (and now immune) and the remaining fraction $1 - \hat{\imath}$ being susceptible. Our scientific question lies in quantifying what the effective reproduction number $R_{t_0}$ equals if all restrictions are lifted at time $t_0$, so $R_{t_0} = R_0(1 - \hat{\imath})$ for a homogeneous population that mixes homogeneously. If $R_{t_0} > 1$ it follows that a new large epidemic outbreak may occur if all restrictions are lifted, as opposed to the case $R_{t_0} \leq 1$ when herd-immunity has been reached (though smaller local outbreaks are still possible).

In the most common COVID-19 scenario that $R_{t_0} > 1$, the minimal amount of preventive measures necessary to avoid a new large outbreak is given by $p_{\text{Min}} = 1 - 1/R_{t_0}$. This amount $p_{\text{Min}}$ is thus a measure of the risk for a new large outbreak. Like the classical formula for minimum vaccination coverage, this expression for $p_{\text{Min}}$ assumes that preventive measures after time $t_0$ are applied uniformly in the population. Of course that is not the case in practice (see Discussion). Nevertheless, the values of $p_{\text{Min}}$ so calculated are still useful for making qualitative comparisons between the level of preventive measures required for populations with different $(R_0, \hat{\imath})$. In the plots in figures 1 and 2 $p_{\text{Min}}$ has been computed as a function of $R_0$ and $\hat{\imath}$, the current immunity level, and is quantified by a heatmap. The left plot in figure 1 is for the main model allowing for heterogeneities with respect to age, social activity and susceptibility, and with disease-induced immunity. The right plot in figure 1 shows the corresponding heatmap when there is no disease-induced immunity but instead immunity comes from vaccinating uniformly in the community (see Discussion on other vaccination policies). Uniform vaccination is equivalent to disease-induced immunity for a model assuming a completely homogeneous community. Figure 2 illustrates the same comparison in one single figure. For three different values of $R_0$, the minimal amount of preventive measures $p_{\text{Min}}$ is plotted as a function of the immunity level, both when immunity comes from disease exposure (solid lines) and when it is achieved by means of vaccination (dashed lines).

The left plot in figure 1 shows that, for a fixed value of $R_0$, the necessary amount of preventive measures needed to avoid a large future outbreak decreases quite rapidly with the amount of disease-induced immunity $\hat{\imath}$, and for $\hat{\imath}$ sufficiently large the colour is deep blue reflecting herd immunity. In

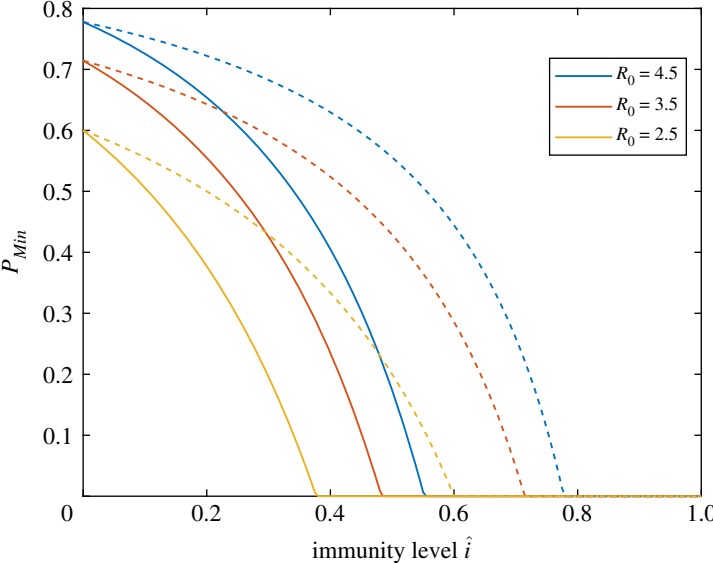

**Figure 2.** Plot of the minimal amount of preventive measures, $p_{\mathrm{Min}}$, as a function of the immunity level $\hat{i}$, for three different values of $R_0$. The solid curve is when $\hat{i}$ comes from disease exposure and the dashed curve when immunity is achieved by vaccination.

the electronic supplementary material, figure S1, we show a similar plot for the model allowing for heterogeneities with respect to age and social activity but not with respect to susceptibility (treated in [1]). The $p_{\mathrm{Min}}$-values for its disease-induced immunity are very similar to those of the present model (left plot of figure 1).

When comparing the effect of disease-induced immunity with the effects of vaccine-induced immunity (assuming uniform vaccination) a difference is clearly observed in each of the two figures. More specifically, for a given $R_0$ and some positive immunity level $\hat{i}$, the necessary amount of preventive measures is substantially higher if immunity comes from vaccination as compared with disease-induced immunity (most easily seen in figure 2). As a numerical illustration, in a region with $R_0 = 2.5$ that has experienced one or several waves in a mitigated situation, resulting in an immunity level of $\hat{i} = 25\%$, the required amount of preventive measures is $p_{\mathrm{Min}} = 29\%$, whereas if instead the same immunity level $\hat{i} = 25\%$ came from (uniform) vaccination, then the necessary amount of preventive measures is $p_{\mathrm{Min}} = 1 - 1/(R_0 \times 0.75) = 47\%$.

An alternative way to compare the effect of disease-induced immunity with vaccine-induced immunity is to compare the corresponding doubling times if all preventive measures are dropped at a time-point when transmission is very low. To illustrate this some further assumptions about the generation time distribution have to be made. These follow from the deterministic SEIR epidemic model and are provided in the electronic supplementary material. If, as above, we consider a region having $R_0 = 2.5$ and immunity level $\hat{i} = 25\%$, then the doubling time for disease-induced immunity equals 12.7 days, whereas it equals 6.6 days if instead the immunity is vaccine-induced. (Electronic supplementary material, figure S2 gives heatmaps for the doubling times as functions of $R_0$ and $\hat{i}$, both when immunity comes from disease exposure and when it is vaccine-induced.) Consequently, if all restrictions were to be lifted the epidemic would start growing much more quickly if immunity came from vaccination. The same qualitative result applies if restrictions are lifted only partially but still below $p_{\mathrm{Min}}$ (the necessary amount of prevention to avoid a new large outbreak).

# 5. Specific regions

We now use estimates of $R_0$ and immunity levels $\hat{i}$ (prior to the second wave) for a few different groups of related regions in order to compare the minimal preventive measures $p_{\mathrm{Min}}$ of regions within each group. The regions that are compared are: Madrid versus Catalonia (containing Barcelona) in Spain, Lombardy (containing Milan) versus Lazio (containing Rome) in Italy, New York State (containing New York City) versus the State of Illinois (containing Chicago), and the three Scandinavian capital regions Stockholm, Copenhagen and Oslo.

As mentioned earlier, the model is a simplification of the real disease spreading situation for COVID-19 by neglecting several heterogeneities (households, spatial aspects, social networks, etc.) and by assuming that earlier preventive measures acted proportionally in the same way between all types of individuals. However, when fitting to data there is also substantial uncertainty in the estimated $R_0$ and $\hat{i}$, so the obtained minimal preventive measures $p_{\text{Min}}$ for different regions are to be interpreted as illustrations rather than exact numbers. Nevertheless, they allow for a qualitative comparison between regions with similar $R_0$ but different immunity levels, as well as a comparison of regions with higher $R_0$ and immunity levels with regions having both lower.

For the sake of illustration, we make the unrealistic assumption that all regions have the same mixing between age groups, the same age structure and the same variation in social activity and variation in susceptibility. For the two empirical measures, age distributions and social mixing patterns, there are of course differences between (and also within!) countries but such differences are of minor order between western countries (e.g. [7]). The only differences considered between compared regions are the overall rate of contacts (measured by $R_0$) and the amount of preventive measures thus leading to different immunity levels $\hat{i}$.

As described above, it is assumed that there is no or low transmission when applying the methodology. However, the results apply also to the situation where there is substantial ongoing transmission, the only difference is that then the issue is not to avoid a new wave, but to make transmission start declining.

In our comparison, we group regions according to similarity in order to make them better comparable in terms of similar mixing, age structure and variable activity and susceptibility, and also so that case fatality data and $R_0$ estimates come from the same data sources/publications within groups. We assume therefore that within each group regions differ only in terms of $R_0$ and immunity level $\hat{i}$, and use estimates of $R_0$ and $\hat{i}$ taken from the same literature source. In the electronic supplementary material, we explain in detail how the estimates are obtained, but briefly it is as follows. The European country-specific $R_0$ estimates are taken from [3] (except for Sweden for which the estimate is taken from its preprint [8], see electronic supplementary material for motivation). To obtain separate estimates of the two Spanish regions we use [9], estimating that Madrid has about 5% higher $R_0$ than Catalonia (a very conservative estimate of the difference). For Italy, Riccardo *et al.* [10] estimates more or less identical (initial) basic reproduction numbers for Lombardy and Lazio, so here we have not distinguished between the two regions. The $R_0$ estimates for New York State and Illinois are taken from [11]. We emphasize that all estimates are based on the original virus strain for COVID-19, and new virus strains seem to have higher $R_0$.

The immunity levels are harder to find estimates of in the literature. For this reason we have used the official number of accumulated case fatalities per 100 000 individuals in the separate regions as of 5 October 2020 (so before the second wave started), and assumed that the infection fatality risk (*ifr*) equals 0.5% (slightly smaller than the earlier *ifr* estimate for China in [12]). By assuming that all infected individuals become immune and that there is no prior immunity, this gives an estimated immunity level. Of course, the *ifr* most likely differs substantially between regions owing to differences in age distribution and health care. Further, the choice to set *ifr* = 0.5% is a rough approximation, as are the assumptions that all infected become immune and that there is no prior immunity. The region-specific immunity levels $\hat{i}$ should hence be seen as illustrations, but their order relations are most likely correct and when this holds true the qualitative comparison statements remain true.

In table 1, the estimated $R_0$ and the disease-induced immunity levels $\hat{i}$ are given first. Then comes $R_{t_0}$ and $p_{\text{Min}}$, computed from the model for the estimated $R_0$ and $\hat{i}$. As a comparison the initial minimal preventive level required to avoid a large outbreak at the start (when $\hat{i} = 0$), $p_{\text{Min}}^{(\text{start})} = 1 - 1/R_0$, is listed, as is the minimal preventive level required to avoid an outbreak if the current immunity level $\hat{i}$ was obtained instead from uniform vaccination, $p_{\text{Min}}^{(\text{Vac})} = 1 - 1/(R_0(1 - \hat{i}))$. The results in table 1 and electronic supplementary material, table SI are of course contingent on the estimates of $R_0$ and $\hat{i}$, which as we indicate above are rather crude. However, by using the software provided with the paper, it is straightforward for interested researchers to explore the effects of alternative estimates of $R_0$ and $\hat{i}$ on the $p_{\text{Min}}$ and doubling times reported in these tables. (Electronic supplementary material, table SI gives the corresponding doubling times if restrictions were lifted.)

It is seen that no studied region has reached close to herd immunity, meaning that none of the regions can lift all restrictions without risking a new large outbreak. By comparing $p_{\text{Min}}$ with $p_{\text{Min}}^{(\text{start})}$, it is further seen that the high levels of required preventive measures at the start of the epidemic have been reduced substantially in regions that have suffered from high transmission during the first epidemic wave. More

**Table 1.** Estimates of $R_0$, COVID-19 fatality rates as of 5 October 2020, the corresponding estimated immunity levels $\hat{i}$, effective reproduction number $R_{t_0}$ and minimal preventive measures $p_{\mathrm{Min}}$. For comparison, the minimal preventive measures needed to avoid a large outbreak at the start, $p_{\mathrm{Min}}^{(\mathrm{start})}$, and the minimal preventive level when the same immunity instead is achieved by uniform vaccination, $p_{\mathrm{Min}}^{(\mathrm{Vac})}$, are listed. See main text for further information.

| region | $R_0$ | deaths/100k | $\hat{i}$ (%) | $R_{t_0}$ | $p_{\mathrm{Min}}$ (%) | $p_{\mathrm{Min}}^{(\mathrm{start})}$ (%) | $p_{\mathrm{Min}}^{(\mathrm{Vac})}$ (%) |
|---|---|---|---|---|---|---|---|
| Madrid | 4.7 | 145 | 29.0 | 2.4 | 58.3 | 78.7 | 70.0 |
| Catalonia | 4.5 | 77.4 | 15.5 | 3.2 | 68.9 | 77.8 | 73.7 |
| Lombardy | 3.4 | 168 | 33.6 | 1.5 | 34.7 | 70.6 | 55.7 |
| Lazio | 3.4 | 16.2 | 3.2 | 3.2 | 68.6 | 70.6 | 69.6 |
| New York State | 4.9 | 169 | 33.8 | 2.2 | 54.4 | 79.6 | 69.2 |
| Illinois | 3.1 | 69.4 | 13.9 | 2.3 | 56.5 | 67.7 | 62.5 |
| Stockholm | 3.9 | 102 | 20.4 | 2.5 | 59.7 | 74.4 | 67.8 |
| Copenhagen | 3.5 | 20.0 | 4.0 | 3.2 | 69.0 | 71.4 | 70.2 |
| Oslo | 3.0 | 11.4 | 2.3 | 2.9 | 65.1 | 66.7 | 65.9 |

specifically, $p_{\mathrm{Min}}$ in Lazio now clearly exceeds that of Lombardy. Catalonia also seems to require slightly more preventive measures than the Madrid region, but the difference is small. New York State required much more preventive measures than Illinois to prevent the first outbreak, but after this outbreak both states required less preventive measures to avoid a second outbreak, and now Illinois requires slightly more than New York. Among the Nordic capital regions, Stockholm had the highest initial minimal preventive measures to avoid an outbreak, whereas after the first wave Copenhagen has highest minimal preventive measures followed by Oslo, but the differences are small.

If instead $p_{\mathrm{Min}}$, values are compared with $p_{\mathrm{Min}}^{(\mathrm{Vac})}$, it is seen that disease-induced immunity plays a more significant roll as compared with immunity achieved by vaccination. In particular, the regions having highest immunity levels (New York State, Madrid and Lombardy) would clearly have larger minimal preventive requirements if immunity had come from vaccination.

# 6. Discussion

The main aim of the paper has been to compare the levels of restrictions needed to avoid new major outbreaks of COVID-19 for different regions having different initial potential ($R_0$) and different current immunity levels $\hat{i}$. Clearly, regions with high $R_0$ that have not yet experienced much spreading need to be most careful, but perhaps more interesting is a comparison between a region with high $R_0$ having experienced much transmission, with another region having smaller $R_0$ but also having lower immunity. The main conclusion from our study is that disease-induced immunity reduces the risk for a large future outbreak substantially more than when the same immunity level is achieved from vaccination. Smaller local outbreaks are possible irrespective of region and are not the focus of the present paper. This result is by no means an argument against vaccination or strong early preventive measures, both of which we strongly support.

In the comparison of different regions, it is seen that the region requiring the highest amount of preventive measures may have switched after the first or subsequent wave from a region with high $R_0$ that has experienced high transmission, to another region having smaller $R_0$ but which has experienced less transmission.

The epidemic model studied allows for individual variation owing to age, social activity and variable susceptibility and infectivity. The age effect is taken from an empirical study. However, the variation owing to social activity and variable susceptibility is chosen arbitrarily but the model for variability lacks any tails and is believed to be less variable than reality. Variable infectivity was shown to have no effect. Of the studied individual variations, social activity plays the biggest role. The explanation of this is that social activity affects both the risk for getting infected *and* the number of people a person infects, so such variations have a dual effect. Variable susceptibility also plays a significant role, and mixing patterns for different age groups has the smallest effect on our results.

Many other heterogeneities are ignored (e.g. households, schools and work places, age differences in susceptibility and infectivity, spatial aspects, travel and commuting) but it is

believed that the effect of adding most such other heterogeneities is that $p_{\text{Min}}$ is shifted close to proportionally.

A greater uncertainty lies in quantifying $R_0$ and the immunity levels $\hat{i}$. The estimated $R_0$ values are based on the early, often irregular, growth of outbreaks, which may be affected by local structures and chance. Further, the estimated immunity levels are based on an infection fatality risk of 0.5% common for all regions (even though the fraction of elderly people varies between the regions). The estimates of $p_{\text{Min}}$ in table 1 should hence only be interpreted as illustrations.

The model does not allow for waning of immunity. The main reasons for this are for mathematical tractability and the fact that our time horizon of interest is smaller than one year. Since the fraction infected even in the worst-hit regions is not more than 30% or so, the community fraction that have first been infected and are by now susceptible again should be very small. Moreover, the qualitative result that individuals with high social activity will be over-represented among the immune is expected to apply also when waning of immunity is taken into account: these individuals will most likely also get reinfected more quickly than others. Recently, new virus strains have emerged for which immunity from the original strain seems to act only partially. Incorporation of such effects is beyond the scope of the present paper but is a highly relevant issue for further study.

An interesting extension could be to consider more realistic preventive measures acting differently between different types of individual, for example, closing restaurants and bars, having greater reduction among socially more active people, and school closure, mainly affecting the younger age groups. The present framework can easily be extended to these situations, the missing information is estimates of how prevention has reduced contacts differently between different pairs of subgroups of individuals. In many countries, risk groups including elderly people have been encouraged to isolate even more. If this was included in the model for prevention, the disease immunity would be distributed even more efficiently in that elderly people with lower mixing rates would have even smaller fractions infected.

It was assumed that vaccinees were selected uniformly from the community. A better vaccination strategy would of course be to select individuals at high risk of getting infected and infecting others, much like how disease-induced immunity is allocated. Such optimal vaccination schemes have received considerable attention in the literature recently (e.g. [13]). One prioritized group in vaccination are health-care workers, and such a strategy might be favourable in this aspect in that they typically have many close contacts, thus increasing the risk to get infected and infecting other. However, another higher prioritized vaccination group are elderly people in order to reduce mortality and morbidity, and such a vaccination policy might be far from optimal if only aiming at reducing transmission, since older people tend to have fewer social contacts. Most likely the combined effect of these vaccination prioritizations is that it reduces transmission *less* than uniform vaccination would do, but on the other hand it reduces mortality and morbidity much more than uniform vaccination would do.

Many countries have a substantial amount of disease-induced immunity from the epidemic waves *and* a usually larger amount of vaccine-induced immunity. The combined effect of these two immunities has not been investigated, but it seems likely that the effect will be that $p_{\text{Min}}$ in this situation lies between $p_{\text{Min}}$ when all immunity is disease-induced and $p_{\text{Min}}$ when all immunity is vaccine-induced, assuming the overall immunity is the same in all three situations.

Recently, new virus strains with higher reproduction numbers are starting to dominate the virus population. The effect of this is that reproduction numbers and herd immunity levels are increased, and that the preventive effect of a given level of immunity is reduced. However, the qualitative difference between disease-induced and vaccine-induced immunity remains.

We conjecture that our two main qualitative results hold true also when adding most of the model extensions mentioned above. These are that the effect of disease-induced immunity on the minimal level of preventive measure is greater than that of vaccine-induced immunity, and that regions having suffered from many infections up until now, may be in a better situation with regard to future outbreaks as compared with other regions with lower $R_0$ but with no or low immunity levels.

Data accessibility. The only 'data' used in the present paper is the contact matrix which is specified on p. 4 in the electronic supplementary material file. The paper uses code to solve the models numerically and to produce the figures and tables. These are explained in the separate electronic supplementary material file titled 'herdcodeRSOS.pdf'. In there, 14 shorter MatLab programs are explained and how they interconnect; all these files are also available in the electronic supplementary material.

Authors' contributions. T.B. initiated and led the project. All authors contributed to the analysis and writing. F.B. did the programming.

Competing interests. We declare we have no competing interests.

Funding. T.B. is funded by the Swedish Research Council (grant 2020-04744). P.T. is funded by the Swedish Research Council (grant 2016-04566).

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
