## [Peer Review File · Royal Society Open Science]

Review History

RSOS-210386.R0 (Original submission)

Review form: Reviewer 1

Is the manuscript scientifically sound in its present form?

Yes

Are the interpretations and conclusions justified by the results?

Yes

Is the language acceptable?

Yes

Do you have any ethical concerns with this paper?

No

Have you any concerns about statistical analyses in this paper?

No

Recommendation?

Accept with minor revision (please list in comments)

Comments to the Author(s)

See attached file (Appendix A).

Review form: Reviewer 2**Is the manuscript scientifically sound in its present form?**

No

Are the interpretations and conclusions justified by the results?

No

Is the language acceptable?

Yes

Do you have any ethical concerns with this paper?

Yes

Have you any concerns about statistical analyses in this paper?

No

Recommendation?

Reject

Comments to the Author(s)

The authors are appreciated for their thorough revision of the manuscript. Fundamentally however, I have trouble with their defenses of their modelling choices. Specifically:

1. In the previous review I pointed out that age distributions differ by a factor of 2 in terms of population over 65. This has not been corrected which means the death rates in Table 1 are incorrect by about that proportion. I do not think this is close enough to justify the statement "the slightly unrealistic assumption that all regions have the same mixing between age groups, the same age-structure". It is a very unrealistic assumption and furthermore, an assumption that is unnecessary to make since the data exist. What is the reason for not using these data? There isn't even need to modify the model if it's not written in a way with the needed flexibility: just multiply the results by a correction factor.
2. Also for the seroprevalence data. The authors state "after (unsuccessful) intensive searching for such data for different regions we gave up". This surprises since I was able to find seroprevalence surveys for most regions. The authors later state "we wanted to have estimates from the same time". This would be best of course, but is not necessary: just run the model up to the date corresponding to that seroprevalence survey. This will give the ratio of diagnosed cases to actual infections up to that point in time (ignoring sero-reversion etc). It is not perfect of course, but is a large improvement on not applying any correction factor.
3. Authors themselves state that they do not believe the R_0 values for Italy, but use it anyway because they do not want to "cherry pick". Some might say that using best available data for each region is not cherry picking but rather good scientific practice! Unless a good review paper has been published one would expect the best data to come from a variety of studies.

4. Most fundamentally: I hugely struggle with "vaccinate anyone at random regardless of risk level or previous infection" as being a scenario worth modelling. Not only is it very unrealistic, but it is unrealistic in a particular way that exaggerates the findings of the study. I do not consider this a reasonable "default" assumption, any more than to say that people become infected at random (which is also a semi reasonable simplification). It would be much fairer to compare random vaccination to random infection, in which case if I understand the study correctly, the two would be equal. Actually, I have not seen any hard evidence for "low, medium, and high" risk individuals (though of course, it seems like it could be true). Yet we know for a fact that high risk individuals are targeted for vaccination (health care workers etc). So it would almost be more realistic to simulate the opposite scenario instead: targeted vaccination and random infections, in which case the findings of the paper would be flipped.

In conclusion I agree with the authors that "several new research papers" could be written on topics related to the study performed here, and I would look forward to those. However I am really struggling to see the value of this particular study, given that it is not just a simplification (which of course would be fine), but systematically at odds with how vaccination is done in the real world.

Decision letter (RSOS-210386.R0)

Dear Dr Britton

The Editors assigned to your paper RSOS-210386 "The risk for a new COVID-19 wave and how it depends on R_0 , the current immunity level and current restrictions" have now received comments from reviewers and would like you to revise the paper in accordance with the reviewer comments and any comments from the Editors. Please note this decision does not guarantee eventual acceptance.

Please submit your revised manuscript and required files (see below) no later than 21 days from today's (ie 01-Jun-2021) date. Note: the ScholarOne system will 'lock' if submission of the revision is attempted 21 or more days after the deadline. If you do not think you will be able to meet this deadline please contact the editorial office immediately.

Please note article processing charges apply to papers accepted for publication in Royal Society Open Science (<https://royalsocietypublishing.org/rsos/charges>). Charges will also apply to papers transferred to the journal from other Royal Society Publishing journals, as well as papers

submitted as part of our collaboration with the Royal Society of Chemistry (<https://royalsocietypublishing.org/rsos/chemistry>). Fee waivers are available but must be requested when you submit your revision (<https://royalsocietypublishing.org/rsos/waivers>).

on behalf of Professor Tim Rogers (Associate Editor) and Mark Chaplain (Subject Editor)
openscience@royalsociety.org

Associate Editor Comments to Author (Professor Tim Rogers):

I see two reasonable responses to the criticisms of Referee 2:

A) Do the extra work to address their concerns with a more carefully parameterised model and more realistic vaccination strategies.

B) Do not do this extra work, but make clear throughout (including in the abstract) that your findings are contingent on these unrealistic assumptions and should be interpreted by readers in light of this.

Reviewer comments to Author:

Reviewer: 1

Comments to the Author(s)

See attached file

Reviewer: 2

Comments to the Author(s)

The authors are appreciated for their thorough revision of the manuscript. Fundamentally however, I have trouble with their defenses of their modelling choices. Specifically:

1. In the previous review I pointed out that age distributions differ by a factor of 2 in terms of population over 65. This has not been corrected which means the death rates in Table 1 are incorrect by about that proportion. I do not think this is close enough to justify the statement "the slightly unrealistic assumption that all regions have the same mixing between age groups, the same age-structure". It is a very unrealistic assumption and furthermore, an assumption that is unnecessary to make since the data exist. What is the reason for not using these data? There isn't even need to modify the model if it's not written in a way with the needed flexibility: just multiply the results by a correction factor.

2. Also for the seroprevalence data. The authors state "after (unsuccessful) intensive searching for such data for different regions we gave up". This surprises since I was able to find seroprevalence surveys for most regions. The authors later state "we wanted to have estimates from the same time". This would be best of course, but is not necessary: just run the model up to the date corresponding to that seroprevalence survey. This will give the ratio of diagnosed cases to actual infections up to that point in time (ignoring sero-reversion etc). It is not perfect of course, but is a large improvement on not applying any correction factor.

3. Authors themselves state that they do not believe the R0 values for Italy, but use it anyway because they do not want to "cherry pick". Some might say that using best available data for each region is not cherry picking but rather good scientific practice! Unless a good review paper has been published one would expect the best data to come from a variety of studies.

4. Most fundamentally: I hugely struggle with "vaccinate anyone at random regardless of risk level or previous infection" as being a scenario worth modelling. Not only is it very unrealistic, but it is unrealistic in a particular way that exaggerates the findings of the study. I do not consider this a reasonable "default" assumption, any more than to say that people become infected at random (which is also a semi reasonable simplification). It would be much fairer to compare random vaccination to random infection, in which case if I understand the study correctly, the two would be equal. Actually, I have not seen any hard evidence for "low, medium, and high" risk individuals (though of course, it seems like it could be true). Yet we know for a fact that high risk individuals are targeted for vaccination (health care workers etc). So it would almost be more realistic to simulate the opposite scenario instead: targeted vaccination and random infections, in which case the findings of the paper would be flipped.

In conclusion I agree with the authors that "several new research papers" could be written on topics related to the study performed here, and I would look forward to those. However I am really struggling to see the value of this particular study, given that it is not just a simplification (which of course would be fine), but systematically at odds with how vaccination is done in the real world.

===PREPARING YOUR MANUSCRIPT===

===PREPARING YOUR REVISION IN SCHOLARONE===

Author's Response to Decision Letter for (RSOS-210386.R0)

See Appendix B.

Decision letter (RSOS-210386.R1)

Dear Dr Britton,

It is a pleasure to accept your manuscript entitled "The risk for a new COVID-19 wave and how it depends on R_0 , the current immunity level and current restrictions" in its current form for publication in Royal Society Open Science.

COVID-19 rapid publication process:

We are taking steps to expedite the publication of research relevant to the pandemic. If you wish, you can opt to have your paper published as soon as it is ready, rather than waiting for it to be published the scheduled Wednesday.

This means your paper will not be included in the weekly media round-up which the Society sends to journalists ahead of publication. However, it will still appear in the COVID-19 Publishing Collection which journalists will be directed to each week (<https://royalsocietypublishing.org/topic/special-collections/novel-coronavirus-outbreak>).

If you wish to have your paper considered for immediate publication, or to discuss further, please notify openscience_proofs@royalsociety.org and press@royalsociety.org when you respond to this email.

on behalf of Professor Tim Rogers (Associate Editor) and Mark Chaplain (Subject Editor)
openscience@royalsociety.org

Appendix A

Re: The risk for a new COVID-19 wave and how it depends on R_0 , the current immunity level and current restrictions

It is great to see that this manuscript that develops a method for determining the risk of new COVID-19 wave has improved. The manuscript, like before, is well-written and kindly see below my comments:

1. The authors define $i(t)$ as the community fraction that cannot get infected- a few currently infectious, but majority having recovered from the disease and now immune (see lines 47 to 49, introduction). From the supplementary material, the age-structure model used is an SEIR model, hence from the basic definition of effective reproduction number: $Rt = R_0x$, x is fraction susceptible. Thus, $x = (1 - p(t))(1 - i(t))$ to the authors – see line 57, introduction section. With the authors' definition, they might be overestimating the effective reproduction number and hence P_{Min} should be smaller than determined. Maybe I am missing something here, as I think the $i(t)$ should just be $r(t)$ – recovered class from the previous wave. Hence, the risk of new wave should include $i(0) = i_{breach}$ as many countries have imposed travel bans, and hotel quarantine is one of the control measures of stopping new introduction. Moreover, the probability of major outbreak can be approximated as: $1 - \left(\frac{1}{R_0}\right)^{i_{breach}}$ if $R_0 > 1$. Just for completeness, $Rt = R_0(1 - p(t))(1 - i_{breach} - r(t))$.
2. In the section, the minimal amount of preventive measures p_{Min} , the authors consider disease-induced immunity versus vaccine-induced immunity for a uniform vaccination strategy, which is similar to the authors homogenous model. How is the uniform vaccination implemented? I think this is not clear to me. In practice, even for uniform vaccination, there will be a percentage of the community that had be infected and immune to the disease before vaccine is introduced. Thus $\hat{i} = 25\%$ should be a combination of percentage immune due to infection and vaccination. With the same level of disease-induced immunity, does additional vaccine coverage, say 10%, require the same level of p_{Min} ? If this does not so, then I think the role of heterogeneity in the application of control measures is well-implicated here.
3. In the section, fitting to specific regions, I think the heading is not appropriate as no fitting was done by the authors. They only extracted fitted results from elsewhere.
4. In line 22, same section, remove the ellipsis.

Appendix B

Dear editors of Royal Society Open Science Journal

We submitted our manuscript entitled “The risk for a new covid-19 wave and how it depends on R_0 , the current immunity level and current restrictions” to Royal Society Open Science in March (after having been transferred from Interface). A few months later we got the manuscript back with suggestions of some minor changes which have now been performed. Below is a detailed description of how each point of the referees and handling editor has been addressed in the revised manuscript.

We hope you find the manuscript interesting and that it has been now been revised satisfactorily, and that you will consider it for publication in your journal.

Kind regards,

Tom Britton (also speaking for my coauthors Pieter Trapman and Frank Ball)

Comments on our revision

Beside addressing the points raised by the referees we have made some minor corrections not worth mentioning in detail. Another small change is that we now have added “prior to vaccination roll-out” in a few places to stress the treated situation (this was the time when the manuscript as written but now this has to be clarified).

Below comes the comments of the handling editor and the two referees, each comment followed by our short response on how the comment has been addressed together with page references to the change(s). In order to distinguish our response from the referee comments ours appear in italics.

Associate Editor Comments to Author (Professor Tim Rogers):

I see two reasonable responses to the criticisms of Referee 2:

A) Do the extra work to address their concerns with a more carefully parameterised model and more realistic vaccination strategies.

B) Do not do this extra work, but make clear throughout (including in the abstract) that your findings are contingent on these unrealistic assumptions and should be interpreted by readers in light of this.

We have chosen to follow track B. In fact we think the unrealistic assumptions had been mentioned in 7 places already prior to this revision: end paragraph of discussion, first paragraph of next section, next paragraph of this section, first paragraph of

"Prevention", 2nd paragraph of "Specific regions", 6th paragraph of same section, 5th paragraph of Discussion. Now we have now added similar comments in the following places: end of abstract, bottom of p8, line 3-6 of 2nd paragraph of "The minimal amount of ...".

Reviewer comments to Author:

Reviewer: 1

Comments to the Author(s)

It is great to see that this manuscript that develops a method for determining the risk of new COVID-19 wave has improved. The manuscript, like before, is well-written and kindly see below my comments:

1. The authors define $i(t)$ as the community fraction that cannot get infected—a few currently infectious, but majority having recovered from the disease and now immune (see lines 47 to 49, introduction). From the supplementary material, the age-structure model used is an SEIR model, hence from the basic definition of effective reproduction number: $R_t = R_0 x$, x is fraction susceptible. Thus, $x = (1 - p(t))(1 - i(t))$ to the authors —see line 57, introduction section. With the authors' definition, they might be overestimating the effective reproduction number and hence P_{Min} should be smaller than determined. Maybe I am missing something here, as I think the $i(t)$ should just be $r(t)$ —recovered class from the previous wave. Hence, the risk of new wave should include $i(0) = i_{breach}$ as many countries have imposed travel bans, and hotel quarantine is one of the control measures of stopping new introduction. Moreover, the probability of major outbreak can be approximated as: $1 - (1/R_0)^{i_{breach}}$ if $R_0 > 1$. Just for completeness, $R_t = R_0(1 - p(t))(1 - i_{breach} - r(t))$.

This is an interesting point raised by the referee. However, in the manuscript we are not considering different measures of avoiding new outbreaks but only studying their combined effect. In fact, when determining p_{min} we first determine R_t "if all restrictions are lifted" (line 7 of section The minimal amount of ...). So R_t is not evaluated assuming quarantine or other prevention. This has now been clarified better by giving the expression for R_t (line 7 of this section).

2. In the section, the minimal amount of preventive measures p_{Min} , the authors consider disease-induced immunity versus vaccine-induced immunity for a uniform vaccination strategy, which is similar to the authors homogenous model. How is the uniform vaccination implemented? I think this is not clear to me. In practice, even for uniform vaccination, there will be a percentage of the community that had be infected and immune to the disease before vaccine is introduced. Thus $\hat{i}=25\%$ should be a combination of percentage immune due to infection and vaccination. With the same level of disease-induced immunity, does additional vaccine coverage, say 10%, require the same level of p_{Min} ? If this does not so, then I think the role of heterogeneity in the application of control measures is well-implicated here.

This is a correct point. However, our comparison consist of comparing disease-induced immunity with the situation that ALL immunity comes from vaccination. This has now been stressed further (line 8, 2nd paragraph of section "The minimal ..."). Further we have added a discussion about the situation mentioned above where some immunity comes from disease and some from vaccination (9'th paragraph of Discussion).

3. In the section, fitting to specific regions, I think the heading is not appropriate as no fitting was done by the authors. They only extracted fitted results from elsewhere.

Correct. We have changed section title.

4. In line 22, same section, remove the ellipsis.

We did not understand this comment and have hence not adjusted anything.

Reviewer: 2

Comments to the Author(s)

The authors are appreciated for their thorough revision of the manuscript. Fundamentally however, I have trouble with their defenses of their modelling choices. Specifically:

1. In the previous review I pointed out that age distributions differ by a factor of 2 in terms of

population over 65. This has not been corrected which means the death rates in Table 1 are incorrect by about that proportion. I do not think this is close enough to justify the statement "the slightly unrealistic assumption that all regions have the same mixing between age groups, the same age-structure". It is a very unrealistic assumption and furthermore, an assumption that is unnecessary to make since the data exist. What is the reason for not using these data? There isn't even need to modify the model if it's not written in a way with the needed flexibility: just multiply the results by a correction factor.

The citation of the referee is wrong: we did nowhere write "slightly unrealistic". We have added additional comments about our simplifications for the illustrations (see comment to the editor).

As explained in the paper our main purpose with the paper is to show a qualitative result: that disease-induced immunity has bigger reducing effect than the corresponding vaccine-induced immunity. Then we illustrate this with some examples. These examples are not claimed to be realistic. In fact, as mentioned in the comment to the editor, we state in more than 10 places that they are examples involving many simplifications and unrealistic assumptions. To make the examples realistic would require much work and much space in the paper, and would switch the focus of the paper in a way the authors would not want.

2. Also for the seroprevalence data. The authors state "after (unsuccessful) intensive searching for such data for different regions we gave up". This surprises since I was able to find seroprevalence surveys for most regions. The authors later state "we wanted to have estimates from the same time". This would be best of course, but is not necessary: just run the model up to the date corresponding to that seroprevalence survey. This will give the ratio of diagnosed cases to actual infections up to that point in time (ignoring sero-reversion etc). It is not perfect of course, but is a large improvement on not applying any correction factor.

See earlier comments. We have added some text referring to these issues at the bottom of p8.

3. Authors themselves state that they do not believe the R_0 values for Italy, but use it anyway because they do not want to "cherry pick". Some might say that using best available data for each region is not cherry picking but

rather good scientific practice! Unless a good review paper has been published one would expect the best data to come from a variety of studies.

This is a matter of taste. Now we are criticized for using the same reference (the most cited Covid paper of all!). We are convinced that the risk of getting criticized would be even bigger if we used different references for different country estimates of R_0 .

4. Most fundamentally: I hugely struggle with "vaccinate anyone at random regardless of risk level or previous infection" as being a scenario worth modelling. Not only is it very unrealistic, but it is unrealistic in a particular way that exaggerates the findings of the study. I do not consider this a reasonable "default" assumption, any more than to say that people become infected at random (which is also a semi reasonable simplification). It would be much fairer to compare random vaccination to random infection, in which case if I understand the study correctly, the two would be equal. Actually, I have not seen any hard evidence for "low, medium, and high" risk individuals (though of course, it seems like it could be true). Yet we know for a fact that high risk individuals are targeted for vaccination (health care workers etc). So it would almost be more realistic to simulate the opposite scenario instead: targeted vaccination and random infections, in which case the findings of the paper would be flipped.

In conclusion I agree with the authors that "several new research papers" could be written on topics related to the study performed here, and I would look forward to those. However I am really struggling to see the value of this particular study, given that it is not just a simplification (which of course would be fine), but systematically at odds with how vaccination is done in the real world.

It is true that in practice the uniform vaccination strategy is not used. Still this method is the most common in comparison studies. We have added text explaining why such comparison is still useful (second from last sentence of "Introduction", line 3-6, 2nd paragraph of section "The minimal amount ...") and also discussed effects of more realistic vaccination schemes (paragraph 9 of Discussion).